# Overexpression of a Grape MYB Transcription Factor Gene *VhMYB2* Increases Salinity and Drought Tolerance in *Arabidopsis thaliana*

**DOI:** 10.3390/ijms241310743

**Published:** 2023-06-28

**Authors:** Chuankun Ren, Zhenghao Li, Penghui Song, Yu Wang, Wanda Liu, Lihua Zhang, Xingguo Li, Wenhui Li, Deguo Han

**Affiliations:** 1Key Laboratory of Biology and Genetic Improvement of Horticultural Crops (Northeast Region), Ministry of Agriculture and Rural Affairs, National-Local Joint Engineering Research Center for Development and Utilization of Small Fruits in Cold Regions, College of Horticulture & Landscape Architecture, Northeast Agricultural University, Harbin 150030, Chinaxingguoli@neau.edu.cn (X.L.); 2Institute of Rural Revitalization Science and Technology, Heilongjiang Academy of Agricultural Sciences, Harbin 150028, China; 3Horticulture Branch of Heilongjiang Academy of Agricultural Sciences, Harbin 150040, China

**Keywords:** grape, *VhMYB2*, salinity, drought, *Arabidopsis thaliana*

## Abstract

In viticulture, the highly resistant rootstock ‘Beta’ is widely used in Chinese grape production to avoid the effects of soil salinization and drought on grape growth. However, the mechanism of high resistance to abiotic stress in the ‘Beta’ rootstock is not clear. In this study, we demonstrated that *VhMYB2* as a transcription factor made a significant contribution to salinity and drought stress, which was isolated from the ‘Beta’ rootstock. The coding sequence of the *VhMYB2* gene was 858 bp, encoding 285 amino acids. The subcellular localization of VhMYB2 was located in the nucleus of tobacco epidermal cells. Moreover, RT-qPCR found that *VhMYB2* was predominantly expressed in the mature leaf and root of the grape. Under salinity and drought stress, overexpressing *VhMYB2* showed a higher resistant phenotype and survival rates in *A. thaliana* while the transgenic lines had a survival advantage by measuring the contents of proline, chlorophyll, and MDA, and activities of POD, SOD, and CAT, and expression levels of related stress response genes. The results reveal that *VhMYB2* may be an important transcription factor regulating ‘Beta’ resistance in response to abiotic stress.

## 1. Introduction

Plants are constantly challenged by biotic and abiotic stresses during their evolution and development [1]. Abiotic stresses, including soil salinization and drought, could affect the normal growth of fruit crops, resulting in the loss of fruit quality and yield [2]. The process of abiotic stress in plants involves a series of regulatory mechanisms in response to adversity [3]. At the molecular level, numerous regulatory and structural proteins play a critical role in the adaptation of plants to abiotic stresses, from the perception of stress signals to the gene expression of stress responses [4]. Multiple transcription factor families are involved in the regulatory network of plant metabolism in response to environmental stresses, for instance, the WRKY, MYB, GRAS, bHLH, ERF, and NAC families [5].

All members of the MYB family of plants contain at least one DNA-binding domain of almost a 53 amino acids structure at the N terminal [6]. Based on the continuous domain number, there are 1R-MYB, 2R-MYB, 3R-MYB, and 4R-MYB four subfamilies [7]. The R2R3-MYB genes arose with roles in plant biological processes, including special metabolism, hormonal signals, and response to stresses [8]. In higher crops, among 126, 89, 94, 196, and 183 R2R3-MYB members have been identified in *Arabidopsis*, rice, pineapple, populus, and chrysanthemums, respectively [9]. Studies on the above species 2R-MYB have found important roles in response to abiotic stresses. For instance, *AtMYB2*, *AtMYB15*, *AtMYB44*, *AtMYB60*, and *AtMYB96* have roles in responses to cold, ABA, salinity, and/or drought in *Arabidopsis* [10]. Relatively, *OsMYB2*, *OsMYB4*, *OsMYB6*, *OsMYB30*, and *OsMYB91* have also been reported to be involved in stress [11]. The function of some R2R3-MYB genes has been investigated in abiotic stress response, but there are still many specific functions of R2R3-MYB transcription factors that have still not been validated. Therefore, further isolation and verification of R2R3-MYB genes are imperative.

The main limiting factors in horticulture are soil salinization and drought, both of which inhibit plant growth and development due to water deficiency [12]. With the perception of stress signals, plants regulate their cellular processes by triggering a network of signaling events that transduce these signals to initiate a domesticated cellular response [13]. Positive and negative stress responses lead to stress tolerance and growth inhibition, respectively, while the latter could be affected by the expression of key genes [14]. *AtMYB2*, a stress-responsive MYB transcription factor, is activated under various abiotic stresses, such as salinity and drought [15]. It has been reported that *OsMYB2* involves the regulation of salinity, drought, and low-temperature stress in rice [16]. *GmMYB76*, a close homolog of *AtMYB2* in soybean, participates in response to high salinity and drought stresses in the transgenic lines, respectively [17].

According to the annotated genome in grapevine (*Vitis vinifera* L.), 134 members of the R2R3-MYB subfamily have been identified in grapes using bioinformatics prediction [18]. Previous studies on the function of the grapevine MYB gene have yielded many unexpected results in the production of plant secondary metabolites [19]. For instance, grapevine *MYB14*, *MYB15*, *MYB40*, and *MYB60* genes regulate the stilbene biosynthesis [20]; *VvMYBA1*, *VvMYBC2-L1*, *VvMYBPA1*, and *VvMYBPAR* genes act as transcriptional activators and repressors to co-modulated anthocyanin and proanthocyanidin biosynthesis [21]; and *VvMYBF1* controls flavonoid synthesis [22]. However, there is inadequate information on the transcriptional regulation of grape R2R3-MYB under salinity stress and drought stress.

The Beta grape is an excellent hybrid rootstock from *Vitis labrusca* x *Vitis riparia* and a native precious germplasm resource of Northeast China with high resistance [23]. Because of the significant positive effect of the Beta rootstock on the accumulation of the total volatile compounds, it is in line with the increasing concern of consumers for high-quality fruit and the effect of rootstock on grape quality [24]. It is worth noting that Beta rootstock will further expand the value of cultivation by enhancing salinity and drought tolerance [25]. Recent studies have shown that *MYB2* functions in abiotic stress resistance in plants, but the research on *MYB2* in Beta has not been an in-depth study. The purpose of this study was to enrich the more valuable functions of the MYB gene and to make a contribution to improving the salinity and drought tolerance of grape rootstocks.

## 2. Results

### 2.1. VhMYB2 Cloning and Phylogenetic Analysis

The gene of *VhMYB2* was homologous to the gene from ‘Pinot Noir’ (VIT_07s0005g01950.t01). The VhMYB2 coding sequence (CDS) of 858 bp was encoded with 285 amino acids as determined with ProtParam (Figure 1). The prediction results of the protein showed that the molecular mass and isoelectric point were 32.25 kDa and 5.38, respectively. The predominated percentage amino acid contents of VhMYB2 protein were Gly (10.2%), Leu (9.5%), Ser (8.8%), and Glu (7.4%), and the hydrophilicity coefficient was −0.767, indicating that VhMYB2 was hydrophilic.

The *VhMYB2* gene included R2 and R3 HTH belonging to the MYB repeat domain in agreement with 13 plant species (Figure 2A). The phylogenetic tree showed the evolutionary relationship of VhMYB2 with other plant species that VhMYB2 was clustered with AtMYB2 in *Arabidopsis thaliana*, so the homology of the two genes was the highest, followed by OsMYB2, OgMYB2, PhMYB2, and PvMYB2 (Figure 2B). The secondary structure of the VhMYB2 protein comprised 37.54% alpha helices, 2.11% beta turns, 2.81% extended strands, and 57.54% random coils (Figure 3A). The VhMYB2 contained two SANT structures located 19–69 aa and 72–120 aa, which indicated that VhMYB2 was a 2R type of the MYB family (Figure 3B). According to the nucleotide sequence of the VhMYB2, the predicted tertiary structure contains two HTH regions by SWISS-MODEL, the same as the number of predicted conserved domains (Figure 3C).

### 2.2. VhMYB2 Localization in Nucleus

To investigate where the *VhMYB2* exercises its function as a transcription factor, we constructed the fusion vector *VhMYB2*-pCAMBIA1300 and transiently expressed it in tobacco leaves. The recombinant vector 35S::VhMYB2-GFP and empty vector (positive control) were co-transformed into *Agrobacterium* GV3101 by the PEG-mediated method and injected into epidermal cells of tobacco leaves with the DAPI, so as to visualize the expression localization of the *VhMYB2* gene. In tobacco epidermal cells, the fluorescence of the positive control was dispersed throughout the cell, whereas the GFP and DAPI fluorescence of the nucleus were overlapped, implying that the VhMYB2 localizes in the nucleus by a confocal scanning microscope (Figure 4).

### 2.3. Expression Level Analysis of VhMYB2 in ‘Beta’

In our experiment, the tissue distribution of the *VhMYB2* transcription in the ‘Beta’ was detected under the control condition by RT-qPCR. The *VhMYB2* showed the most prominent expression in the mature leaf, followed by the root, while the low expression was in the stem and young leaf (Figure 5A).

The expression of the *VhMYB2* was induced in response to 100 mM NaCl, 6% PEG6000, 4 °C/37 °C, or 100 μM ABA stress treatments. At different time points within 24 h of processing, the *VhMYB2* expression levels showed similar peaks in the mature leaf and root. Interestingly, at high concentrations of NaCl treatment for 4 h and 8 h in the mature leaf and root, the peak of the *VhMYB2* expressions was 11.9 times and 8.1 times higher than those of the untreated groups, respectively (Figure 5B). For the PEG6000 treatment for 12 h and 4 h, the peak of the *VhMYB2* in the mature leaf and root was 9.8 times and 9.2 times higher than those of the untreated groups, respectively (Figure 5C). These results suggested that the *VhMYB2* expression levels in the mature leaf and root were dramatically induced when responding to the NaCl and PEG6000 treatments compared with the other treatments.

### 2.4. VhMYB2 in Transgenic A. thaliana Enhanced Salinity Tolerance

To probe the *VhMYB2* function in salinity and drought, the overexpressing *VhMYB2* and unloaded line (UL) in T_1_ were screened for vector resistance. The transgenic *A. thaliana* of the *VhMYB2* overexpressing in T_2_ was analyzed by RT-qPCR, with the WT and UL plants as the control (Figure 6A). The experiment demonstrated that there was no expression of *VhMYB2* in the control, but there were different expression levels in the transgenic lines, implying that the target gene could be expressed normally in the transgenic *A. thaliana*. Finally, three transgenic lines (L1, L6, and L7) with a higher expression were selected from eight transgenic lines (L1–L8) and cultured into the T_3_ generation.

In order to elucidate the role of the *VhMYB2* on *A. thaliana* under high salinity stress, salinity tolerance experiments of the transgenic (L1, L6, L7), UL, and WT plants were carried out. For the phenotyping, the transgenic (L1, L6, L7), UL, and WT plants were grown in normal conditions for three weeks at 25 °C, after which they were transferred to a nutrition bowl that contained a 200 mM NaCl solution and grown for 10 days. Under normal conditions, the phenotype of the transgenic plants (L1, L6, L7) was similar to that of the UL and WT seedlings (Figure 6B). After being treated with the 200 mM solution for 10 days, the leaves of the transgenic plants (L1, L6, L7), UL, and WT plants were significantly yellow, especially those of the UL and WT plants (Figure 6B). The survival rates of all the lines under the control condition were similar, but after high salinity recovery, the survival rates of the overexpression lines (L1, L6, L7) were 78%, 82%, and 79% while those of the UL and WT plants were 36% and 34% (Figure 6C). This difference is definitive that the overexpression lines have a higher survival rate compared with UL and WT in response to salinity treatment.

Moreover, the physiological indexes of the transgenic lines (L1, L6, L7), UL, and WT lines under the control and high salinity stress were determined. The physiological indexes of all the plants under the control condition had no obvious difference. On the contrary, the activities of the CAT, SOD, and POD of the transgenic lines under salinity stress were obviously high; meanwhile, the contents of the proline, chlorophyll, and MDA of the transgenic lines, rather than the WT and UL lines, also varied widely (Figure 7A–F). The experiment suggested that the overexpression of *VhMYB2* may enhance the tolerance to the transgenic lines under high salinity stress.

### 2.5. VhMYB2 in Transgenic A. thaliana Activated Salinity Tolerance-Related Genes

In order to elucidate the regulatory mechanisms of the *VhMYB2* in response to salinity stress, the expression of the salinity-responsive genes, including *AtSOS2* (NM_122932.5), *AtSOS3* (NM_122333.6), *AtSOS1* (NM_126259.4), and *AtNHX1* (NM_122597.3), were investigated in the *VhMYB2* transgenic plants. The RT-qPCR analysis revealed that there were no obvious expression differences of *AtSOS2*, *AtSOS3*, *AtSOS1*, and *AtNHX1* among the transgenic, UL, and WT lines under the absence of stress factors, yet the above genes expression of the transgenic lines were obviously higher than the WT and UL lines under salinity stress (Figure 8A–D). These changes suggested that the *VhMYB2* transgenic lines possibly activated the expression of the salinity-related genes, enhancing the tolerance under high salinity stress.

### 2.6. VhMYB2 in Transgenic A. thaliana Enhanced Drought Tolerance

The *Arabidopsis* lines of the transgenic (L1, L6, L7) together with UL and WT were used to verify the *VhMYB2* function in drought tolerance. Those plants grew well under the control condition and had no obvious phenotypic change during three weeks. When there was a two-week water deprivation condition at the seedling stage, the leaves of the UL and WT lines showed symptoms of stunted growth and wilting. Unlike the WT and UL lines, most leaves remained green and there were slightly wilting symptoms in the *VhMYB2* overexpression *Arabidopsis* transgenic plants at this time (Figure 9A). After three days of re-watering, 81%, 78%, and 82% of the *VhMYB2* transgenic plants (L1, L6, L7) exhibited normal growth with higher survival rates; however, the majority of the UL and WT plants died (Figure 9B). The experiment indicated that the survival rates of the *VhMYB2* overexpression lines were a lot higher than the WT and UL lines when growing in a nutrition bowl without water.

Furthermore, the physiological indexes of the transgenic lines (L1, L6, L7), UL, and WT lines under the control and drought stress were determined. The physiological indexes of all the plants under the control condition had no obvious difference. On the contrary, the activities of the CAT, SOD, and POD of the overexpression lines under drought stress were obviously high; meanwhile, the contents of the proline, chlorophyll, and MDA of the transgenic lines, rather than the WT and UL lines, also varied widely (Figure 10A–F). Taken together, the experiment suggested that the overexpression of the *VhMYB2* may enhance the tolerance to the transgenic lines under drought stress.

### 2.7. VhMYB2 in Transgenic A. thaliana Activated Drought Tolerance-Related Genes

To detect the drought-responsive genes of the *VhMYB2* transgenic lines, we chose ABA signal transduction *AtSnRK2.6* (NM_119556.3), ABA biosynthetic genes *AtNCED3* (NM_112304.3), proline synthesis gene *AtP5CS1* (NM_129539.2), and the ROS scavenging gene *AtCAT1* (NM_101914.4) for RT-qPCR. The quantitative data exhibited that above these genes in the *VhMYB2* transgenic lines, the UL and WT lines had no obvious expression differences under the absence of stress factors, yet the above gene expression of the transgenic lines was obviously higher than the WT and UL lines under drought stress (Figure 11A–D). These changes indicated that the transgenic lines possibly activated the expression of the drought-related genes, responding to enhancing the tolerance under drought stress.

## 3. Discussion

Beta, a grape rootstock, is widely used in viticulture with a positive impact on the grapevine response to abiotic stresses [23]. Facing the challenges of future modern grape cultivation, the exploitation and application of grape rootstock genotypes is a new breeding opportunity [26]. Many studies have demonstrated that MYB transcription factors regulate abiotic stress, yet the regulatory mechanism of the MYB transcription factor in grape rootstocks is little known [27]. Here, we used hybrid Beta as experimental materials from the grape gene bank, identified as an R2R3-MYB sequence according to the AtMYB2 protein sequence homology. A novel gene, *VhMYB2*, was cloned using primers designed with snapgenes, containing 858 bp nucleotide encoded with 285 amino acids. The phylogenetic tree analysis of the other MYB2 proteins and *Arabidopsis* MYB gene family showed that *VhMYB2* and *AtMYB2* had a higher phylogenetic relationship (Appendix A). Many studies have shown that plant transcription factors mainly function in the nucleus by observing onion epidermal cells, *Arabidopsis* protoplast cells, and epidermal tobacco cells with laser-scanning confocal microscopy [28,29,30,31]. A 35S::VhMYB2-GFP vector was injected into the tobacco epidermis and, 48 h later, the expression position of the gene was confirmed in the nucleus by microscopy (Figure 4). The result is also the same as a previous study of MYB108 distribution in grapes [32].

MYB transcription factors, particularly R2R3-MYB, have been extensively studied during plant growth [33]. To date, these genes have been validated for abundant functions, such as the regulation of cell morphogenesis, the regulation of secondary metabolism, and the response to stress [34]. In *Arabidopsis*, it has been proved *AtMYB16* is not only preferentially functioned in the stomatal lineage ground cells, but also participates in regulating the process of the morphological differentiation of pavement cells and guard cells [35]. Recent studies pointed out that the *Arabidopsis thaliana* genes *MYB99*, *MYB24*, and *MYB21* participate in the secondary metabolic activities of *Arabidopsis thaliana* by regulating the phenylpropane metabolic pathway [36]. The *AtMYB30* transcription factor positively regulates plant tolerance in *A. thaliana* by balancing ROS signaling and systemic adaptive responses to bloom stress [37].

Studies have shown that *AtMYB2* and *OsMYB2* work mainly at the leaves and roots [15,16] (The Bio-Array Resource for Plant Functional Genomics, http://bar.utoronto.ca/, accessed on 7 February 2022). Similarly to *AtMYB2* and *OsMYB2*, our survey of the *VhMYB2* expression in grape tissues implied that the gene worked primarily in mature leaves and roots (Figure 5A). The RT-qPCR showed that high-salt, drought, exogenous ABA, cold-, and hot-induced *VhMYB2* expression varied with the time of treatment. In salinity and drought stress, the *VhMYB2* expressions in the mature leaf and root were up-regulated while the root was more obvious; however, the cold and hot did not have a more sensitive effect on the accumulation of the *VhMYB2* in the mature leaf and root (Figure 5). Previous studies have proved that *GmMYB92* and *CmMYB2* also have higher expression levels in roots or leaves, respectively [17,38]. This implied that after the hybrid grape *V. labrusca* × *V. riparia* ‘Beta’ was in a growth environment with 200 mM NaCl, the root may first respond to salinity stress.

To further understand the function of the *VhMYB2* gene, we constructed *VhMYB2*-overexpressing *Arabidopsis* lines and tested their physiological responses to salinity and drought stress. When plants were subjected to a complex environment leading to ion balance, the ability to maintain osmotic homeostasis after feeling the pressure was an important part of their adaptation to the environment [39]. The regulation of chlorophyll and proline in cells helped to maintain the water potential and bulging pressure of cells, thus improving salinity and drought tolerance [40]. The measurements of chlorophyll can detect cellular responses to the degrees of salinity and drought stresses in plants [41]. Proline was one of the key organic osmolytes regulating the relationship between the cytoplasmic and vacuolar water in plants in response to oxidative stress, including salinity and water shortage [42]. The experiment showed that the transgenic lines accumulated more chlorophyll and proline than the UL and WT lines under salinity and drought treatments, manifesting that the overexpression of *VhMYB2* may be responsible for salinity and drought tolerance.

Salinity and drought are the main abiotic stresses in the world, which can induce the accumulation of ROS and MDA to evaluate the degree of plasma membrane damage and the stress tolerance of plants [43]. The antioxidant system had a complex and multi-layer network, which can induce the antioxidant enzymes CAT, SOD, and POD to resist the harmful reaction of mainly ROS and maintain the homeostasis of the cells [44]. The overexpressed *LcMYB2* promoted the enzyme activity of the SOD but reduced the MDA content so as to reduce the harm to the transgenic lines under various adverse growth conditions [45]. Compared with plants grown under normal conditions, the overexpression of *SbMYB2* or the *SbMYB7* transgenic lines under drought and salt stress showed higher activities of CAT, SOD, and POD [46]. Consistent with our experimental results, there was no significant difference in the physiological indexes of each plant under the control condition; however, the higher activities of CAT, SOD, and POD, and the lower content of the MDA of the transgenic lines were different from those of the WT and UL lines under salinity and drought stress. Collectively, these results demonstrated that the salinity and drought tolerance of the *VhMYB2* transgenic lines were partially related to the lower MDA content, higher activities of CAT, SOD, and POD, and higher contents of proline and chlorophyll.

The expression of the response genes corresponding to the above physiological indexes and enzyme activities were also related to salinity and drought tolerance [47]. For instance, the *FvMYB82* and *MbMYB108* overexpression in transgenic *A. thaliana* influenced its tolerance to salinity and drought stress by activating the expression of stress response genes [48,49]. Consistent with our study, the expression levels of *AtP5CS*, *AtCAT1*, *AtNCED3*, and *AtSnRK2.6* in plants overexpressing *VhMYB2* were higher than those in UL and WT under drought stress (Figure 8). The P5CS involved in the first step of proline biosynthesis under drought stress, and the up-regulation of the *P5CS* gene in the 35S::VhMYB2 lines, were consistent with the findings on *OsMYB6* and *GmMYB76* [11,17]. It has been reported that the expression of *AtP5CS* contributes to accumulating the proline contents and improving the tolerance in the transgenic lines [50]. In addition, *VvWRKY28* may activate *AtP5CS*, thereby promoting proline biosynthesis [51].

ABA is a key mediator regulating plant responses to abiotic stresses, and ABA synthesis-related and signal transduction genes play an important role [52]. ABA biosynthesis depends on the stress induction of the *NCED3* gene, which is a rate-limiting step in the ABA pathway and regulates ABA accumulation [53]. The overexpression of *AtNCED3* and *OsNCED3* in the transgenic lines could strengthen drought tolerance and add ABA content, which indicates the conserved importance of *AtNCED3*/*OsNCED3* in ABA biosynthesis in response to drought stress [54]. ABA signaling transduction depends on SnRK2s protein kinase autophosphorylation and the activation of downstream receptors, thereby responding to plant drought stress [55]. Under drought stress, *AtSnRK2.6* is involved in ABA-mediated stomatal closure, thereby improving drought tolerance [56]. This suggests that *VhMYB2* could mediate the drought response by participating in the ABA pathway. Moreover, the overexpression of *GmMYB118* can propel the expression levels of *AtP5CS1*, *AtNCED3*, *AtRD29A*, and *AtCOR15*, which improves the drought stress tolerance of transgenic *A. thaliana* [57].

The SOS (salt-overly-sensitive) pathway is regarded as a key mechanism to salinity tolerance by exporting excess Na^+^, which includes three SOS genes (*SOS1*, *SOS2*, and *SOS3*) [58]. In *A. thaliana*, *SOS2* is the key regulatory step that underpins the formation of the protein kinase complex *SOS2*-*SOS3*, which activates *SOS1* to regulate salinity tolerance [59]. The plasma membrane Na^+^/H^+^ antiporter *SOS1* has been shown to improve salinity tolerance in *Arabidopsis* [60]. The tonoplast Na^+^/H^+^ antiporters NHXs in plants actively transport Na^+^ to the vacuoles and are regulated by the SOS pathways. Consistent with the *AtNHX1* report, *NHX1* also improved salinity tolerance in rice, tomatoes, and apples, respectively [61]. This study revealed that the relative expression of the four genes related to salinity stress was significantly increased in the overexpressed *VhMYB2* lines (Figure 11). It has been reported that *MYB42* plays a positive role in regulating the SOS pathway by binding to the *SOS2* promoter, thereby mediating salinity tolerance in *Arabidopsis* [62]. On the contrary, the *AtMYB73* negatively modulates the SOS pathway, resulting in salinity sensitivity [63]. In contrast, these data imply that the overexpressed *VhMYB2* lines may promote salinity tolerance, thereby positively regulating *AtNHX1* and *AtSOS1/2/3* and keeping the intracellular Na^+^ equilibrium.

In summary, we have developed a potential model to describe the role of *VhMYB2* in salinity and drought stresses according to the experiment and previous studies (Figure 12). Under salinity stress, *VhMYB2* promotes the downstream-salinity-stressed-related gene expression manifests that *SOS2*-*SOS3* mediates *SOS1* and *NHX1* to enhance the salinity tolerance. Under drought stress, *VhMYB2* promotes the downstream-drought stressed-related genes *NCED3, SnRK2.6*, *CAT1*, and *P5CS*, thereby improving drought tolerance.

## 4. Materials and Methods

### 4.1. Plant Materials and Treatment Conditions

The hybrid grape *V. labrusca* x *V. riparia* ‘Beta’ plants were obtained from Northeast Agricultural University, Harbin, China. The tissue culture plantlets of Beta (*V. hybrid*) were cultured on bud induction medium (Murashige and Skoog + 0.2 mg/L indole-3-butyric acid + 0.5 mg/L 6-benzylaminopurine) for 30 days to expand the tissue culture seedlings or transfer to rooting medium (MS + 0.2 mg/L IBA + 0.1 6-BA), which were kept in an incubator (22 °C with a cycle of 16 h light/8 h dark) at the College of Horticulture, Harbin, China. The wild-type *A. thaliana* seeds from our research group were Colombian ecotypes. The seedlings of *Arabidopsis* were cultured in an incubator under the same growing conditions as ‘Beta’.

The grape seedlings with numerous roots were transferred to Hoagland solution while 50 seedlings with well-developed roots, stems, and leaves were selected for stress treatments. The leaves (young/mature), stems, and roots of ‘Beta’ were taken for the tissue-specific expression analysis of the *MYB2* gene. To test the expression pattern of *VhMYB2* to drought, salt, low/high temperature, and ABA stress tolerances, the seedlings growing in Hoagland solution were treated with 100 mM NaCl, 6% PEG6000, 4 °C/37 °C or 100 μM ABA, respectively. ‘Beta’ samples treated with abiotic stresses were collected at six-time intervals (0, 1, 2, 4, 8, 12, and 24 h). All samples were quickly frozen with liquid nitrogen and stored at −80 °C for later analysis.

### 4.2. Isolation and Cloning of VhMYB2

Total RNAs were extracted from the samples and were used for a template to synthesize cDNA by Omega Bio-Tek company kits, which were HP Plant RNA Kit and M-MLV First Station cDNA Synthesis Kit, respectively. With reference to the *VvMYB2* nucleotide sequence (NM_001280988.1) in the grape (*Vitis vinifera* L.) genome assembly, gene-specific primers were designed using Snapgene 6.0 software (Appendix A). The target gene was amplified using C1000^TM^ Touch Thermal Cycle PCR instrument of BIO-RAD. The obtained target fragments were detected with 1% TAE solution and purified, then ligated with the T5 cloning vector and sequenced (BGI, Beijing, China) [64].

### 4.3. Subcellular Localization of VhMYB2

Using the full-length *VhMYB2* sequence and the restriction sites incorporated in the pCAMBIA1300-GFP vector (Bioon, Shanghai, China), *BamH*I and *Sal*I were selected as the restriction sites, and specific primers (Appendix A) including the restriction sites were designed. The pCAMBIA1300-GFP vector was linearized by *BamH*I and *Sal*I digestion while the homologous arms were attached to both ends of the target fragment by PCR amplification with double-digested primers. The digested DNA fragment and the vector were ligated to construct the *VhMYB2*-GFP vector. The target fragment with homologous arm was connected with the linearized vector. Meanwhile, the fusion *VhMYB2*-pCAMBIA1300-GFP expression vector and empty vector were co-introduced into tobacco leaf using *Agrobacterium tumefaciens* strain GV3101, respectively, with the transformation of empty vector as positive control [65]. After culturing 48 h in weak light, the nuclei of tobacco leaf cells were stained by DAPI while the distribution of *VhMYB2*-GFP and control plasmids were observed by a confocal fluorescence microscope (LSM 900, Precise, Beijing, China).

### 4.4. Sequence Analysis of VhMYB2

The amino acid sequence of *VhMYB2* was analyzed using NCBI databases (https://www.ncbi.nlm.nih.gov/, accessed on 7 February 2022) and TBtools software 1.0 [66]. Total average hydrophilic coefficients, theoretical isoelectric point, and relative molecular mass of VhMYB2 proteins were predicted using the ExPASy ProtParam tool (https://www.expasy.org/resources/protparam, accessed on 7 February 2022). The tertiary structure with high score was obtained by inputting VhMYB2 protein sequence on SWISS-MODEL website (https://swissmodel.expasy.org/, accessed on 7 February 2022). Based on sequence similarity to the VhMYB2 protein, the similar sequences from BLAST were aligned. As shown in Figure 2B, the gene accession numbers are indicated on a phylogenetic tree for the homologous proteins constructed using MEGA7 and modified by Fig Tree v 1.4.

### 4.5. Expression Analysis of VhMYB2

*VhMYB2* expression was measured by RT-qPCR following the method of Han et al. [67]. The *Actin* gene (XM_002282480) was selected as the reference gene with stable expression. Based on conserved regions of *VhMYB2* and *Actin* sequences, the RT-qPCR primers were designed with high specificity (Appendix A). The RT-qPCR analysis was performed using SYBR^®^ Green Realtime PCR Master Mix (TaKaRa, Beijing, China). The parameters set for the whole quantitative reaction were as follows: pre-denaturation (95 °C, 30 s), then denaturation (95 °C, 5 s), and annealing (60 °C, 10 s) for 40 cycles. The 2^−ΔΔ*C*t^ method was used to analyze the relative expression of *VhMYB2* [68]. The experiment was carried out with three technical replicates.

### 4.6. Vector Preparation of Transgenic Arabidopsis

Homology arms and restriction sites (*Kpn*I and *Xba*I) were designed according to the cloned coding sequences (CDS) and plasmid pCAMBIA2300 sequences with the CaMV35S promoter. The overexpression vector *VhMYB2*-pCAMBIA2300 was constructed by homologous recombination primers and system (Appendix A). To obtain *VhMYB2* transgenic *A. thaliana*, the empty vector (pCAMBIA2300) and the overexpression vector (*VhMYB2*-pCAMBIA2300) were introduced into GV3101 and transformed into *Arabidopsis* Columbia ecotype by inflorescence permeation methods [69]. The *A. thaliana* seeds from inflorescences were screened on medium (1/2 MS + 50 mg/L Kanamycin). The T_2_ transgenic *A. thaliana* was cultured for quantitative analysis and the T_3_ homozygous lines were obtained by selecting the three T_2_ high-expression transgenic lines (L1, L6, and L7).

### 4.7. Analysis of Related Physiological Indexes in VhMYB2-OE A. thaliana

Before treatment with salinity stress (200 mM NaCl) and drought stress, T_3_ transgenic *A. thaliana* lines (L1, L6, and L7), unload line (UL), and the wild-type (WT) were seeded and cultured in light incubator with alternating light and dark (16 h/8 h) at 22 °C for three weeks. The T_3_ transgenic lines, UL, and WT were irrigated with salt solution daily for 10 days. The drought group was treated for 14 d and later cultured with re-watering conditions for 3 days. During whole stress treatment and recovery treatment, the morphological changes of plants were recorded and observed. Moreover, samples (L1, L6, L7, UL, and WT) were collected to determine the contents of chlorophyll (Chl), proline (Pro), and malondialdehyde (MDA) using the spectrophotometer, acid ninhydrin method, and thiobarbituric acid method, respectively [70]. Similarly, the activities of peroxidase (POD), surperoxide dismutase (SOD), and catalase (CAT) were determined by the guaiacol method, tetranitrotetrazolium blue reduction method, and ultraviolet spectrophotometry, respectively [71].

### 4.8. Expression Analysis of Related-Stress Genes in VhMYB2-OE A. thaliana

Total RNAs extracted from the lines were used for a template to synthesize cDNA. The expression levels of stress-related genes, including ABA-dependent response genes (*AtNCED3* and *AtSnRK2.6*), ABA-independent response genes (*AtP5SC* and *AtCAT1*), and salinity-stress-responsive genes (*AtSOS2*, *AtSOS3*, *AtSOS1*, and *AtNHX1*) were determined with RT-qPCR. The RT-qPCR primers of related stress and the internal reference genes (*AtACTIN2*) were listed in Appendix A. The RT-qPCR program was consistent with Section 2.5.

### 4.9. Statistical Analysis

Three biological replicates were performed for each sample, treatment, and assay, which were ultimately presented as mean ± standard errors (SE). Significant differences (* *p* ≤ 0.05 and ** *p* ≤ 0.01) were assessed through one-way ANOVA tests with IBM SPSS Statistics 26.0.

## 5. Conclusions

In this study, a grape rootstock ‘Beta’ (*Vitis labrusca* × *Vitis riparia*) MYB transcription factor, *VhMYB2*, was isolated and characterized. The expression levels of *VhMYB2* were induced by salinity and drought stresses in the root and mature leaf of the grape. The overexpression of the *VhMYB2* rendered *A. thaliana* more tolerant to salinity and drought environments with changes in the related physiological indicators. The expression of a single MYB-transgene in the transgenic lines was not enough to cause the expression of the downstream genes associated with the stress response, which was observed when expressed in the absence of the stress factors. This MYB-transgene significantly contributed to the increase in stress tolerance in transgenic plants during the stress induction of other yet unidentified genes. This study is a significant step for the molecular breeding of grape rootstock improvement.

## Figures and Tables

**Figure 1 ijms-24-10743-f001:**
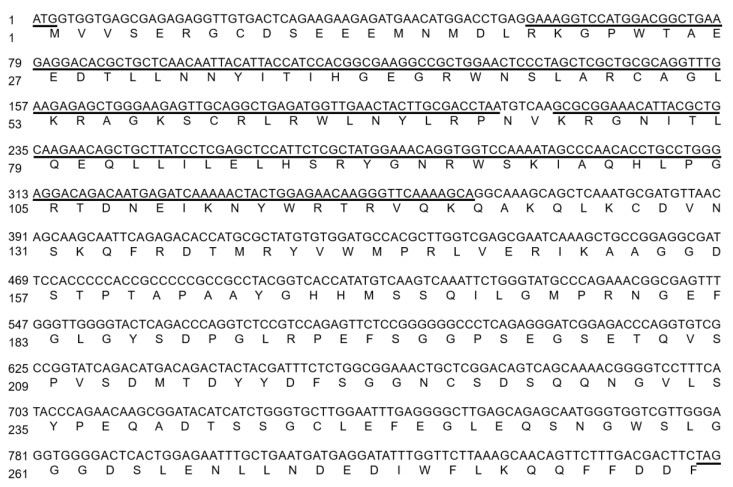
Nucleotide and deduced amino acid sequences of VhMYB2. Amino acid sequences that are conserved members of the MYB family are underlined by black lines, which consist of R2 and R3 conserved domains.

**Figure 2 ijms-24-10743-f002:**
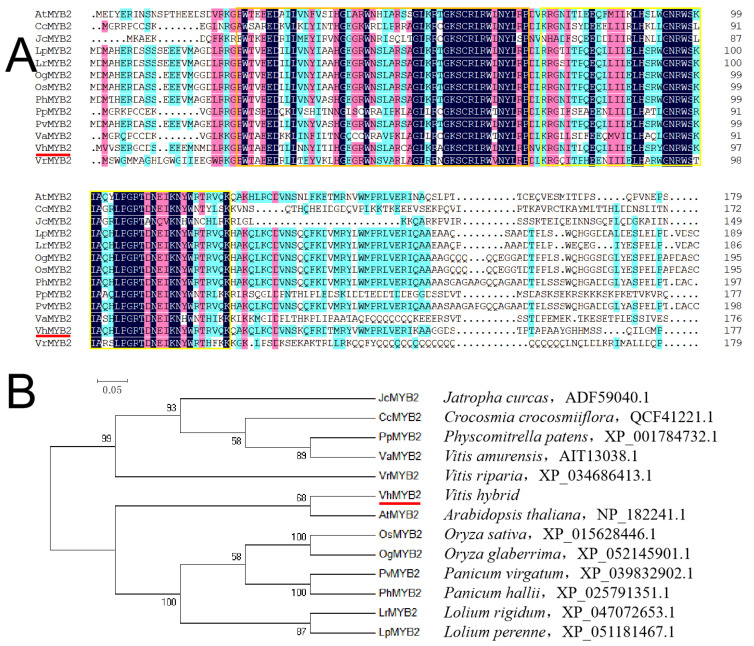
Sequence analysis of VhMYB2. (**A**) Alignment of VhMYB2 and MYB2 proteins of other species. The orange and yellow boxes indicate R2 and R3 domains, respectively. (**B**) Phylogenetic tree of VhMYB2 and MYB2 proteins of other species. The red lines indicate the VhMYB2 protein.

**Figure 3 ijms-24-10743-f003:**
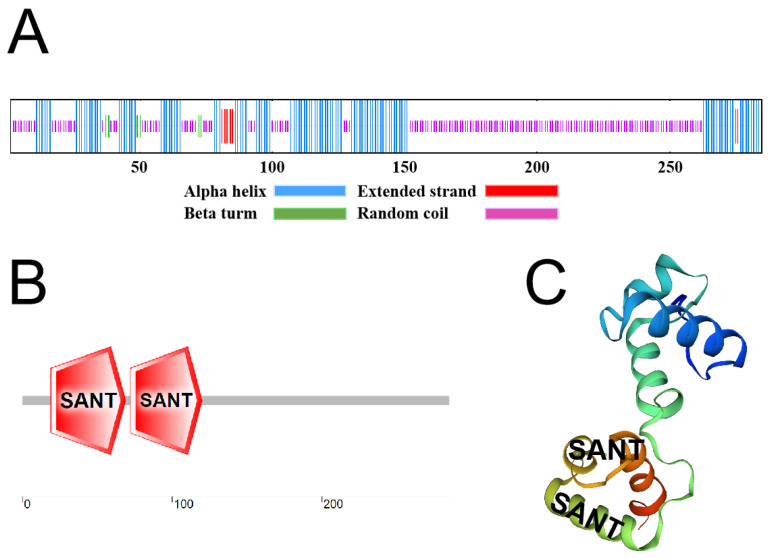
Structure analysis of VhMYB2 protein. (**A**) Secondary structure analysis of VhMYB2. (**B**) Conservative domain analysis of VhMYB2. (**C**) Spatial structure of VhMYB2.

**Figure 4 ijms-24-10743-f004:**
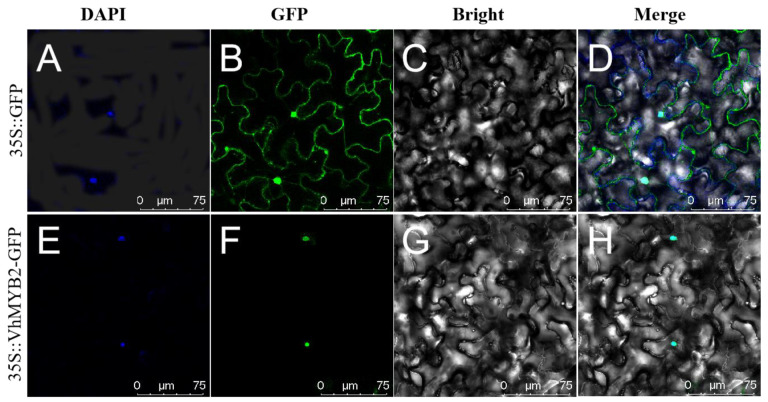
Subcellular localization of VhMYB2 protein in tobacco plant tissue. (**A**–**D**) Location of empty vector 35S::GFP (positive control) in tobacco epidermal cells. (**E**–**H**) Location of recombinant vector 35S::VhMYB2-GFP in tobacco epidermal cells. From left to right, the micrographs showed DAPI, GFP, Bright, and Merge of three illuminations. Bar = 75 μm.

**Figure 5 ijms-24-10743-f005:**
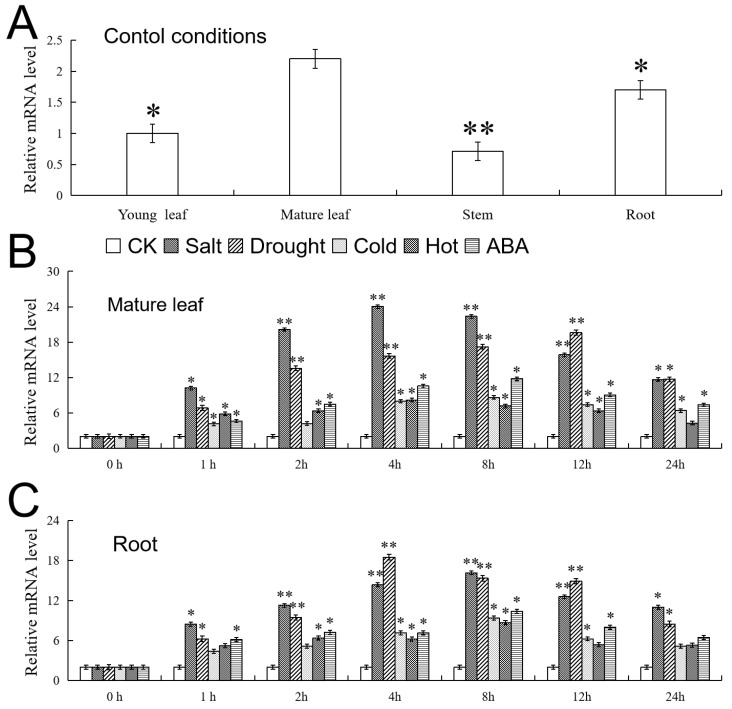
Expression patterns of *VhMYB2* gene in “Beta” plants. (**A**) *VhMYB2* relative expression of young leaf, mature leaf, stem, and root. (**B**) *VhMYB2* relative expression of mature leaf treated with 6% PEG6000, 100 mM NaCl, 4 °C/37 °C, or 100 μM ABA treatments, respectively. (**C**) *VhMYB2* relative expression of root treated with 6% PEG6000, 100 mM NaCl, 4 °C/37 °C, or 100 μM ABA treatments, respectively. Data represent the average of three repetitions. Asterisks above the error bars indicate a significant difference between the treatment and control (* *p* ≤ 0.05; ** *p* ≤ 0.01).

**Figure 6 ijms-24-10743-f006:**
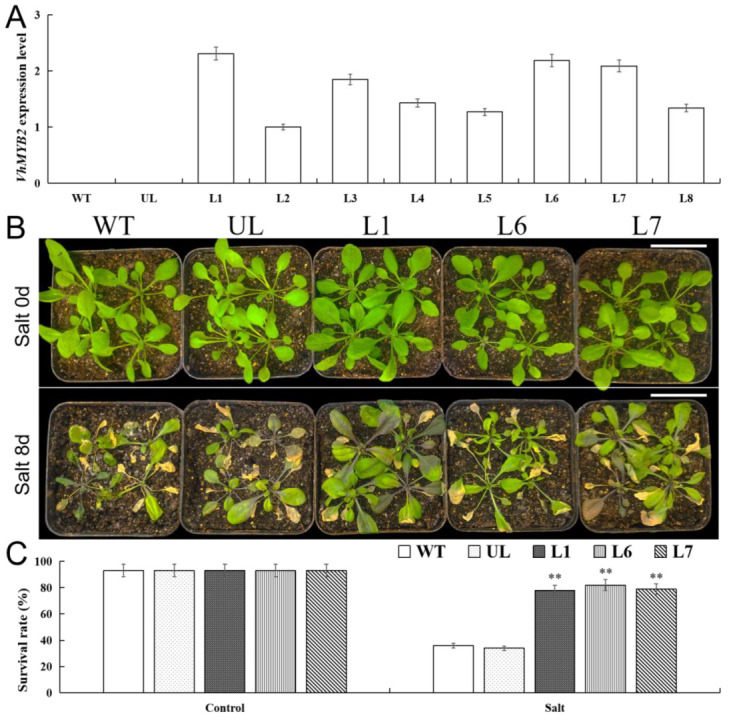
35S::*VhMYB2* transgenic *A. thaliana* plants. (**A**) *VhMYB2* gene expression of WT (wild-type), UL (unload line), and L1–L8 (transgenic lines). (**B**) Phenotypes of WT, UL, and transgenic lines (L1, L6, and L7) with high expression. Scale white bar = 3.5 cm. (**C**) Survival rates of *A. thaliana* under salinity stress. Asterisks indicate significant difference compared to WT (** *p* ≤ 0.01).

**Figure 7 ijms-24-10743-f007:**
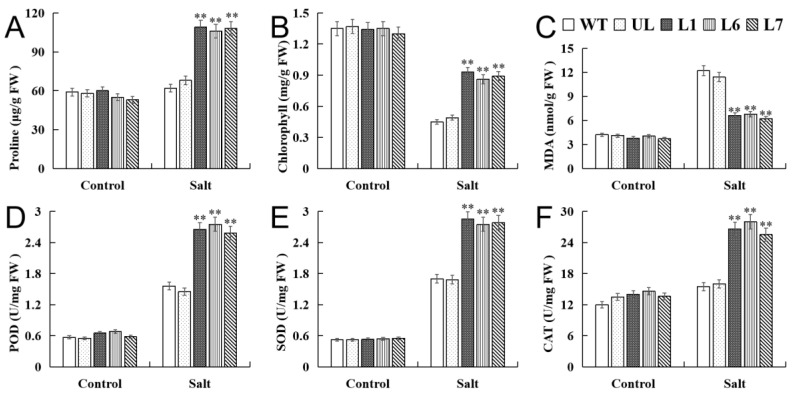
Analysis of proline (**A**), chlorophyll (**B**), MDA (**C**) POD (**D**), SOD (**E**), and CAT (**F**) in WT and 35S::*VhMYB2* transgenic *A. thaliana* plants. These were measured at 0 and 10 d after salinity treatment. Asterisks above the error bars indicate a significant difference between transgenic and WT lines (** *p* ≤ 0.01). The level of each index in WT was used as control.

**Figure 8 ijms-24-10743-f008:**
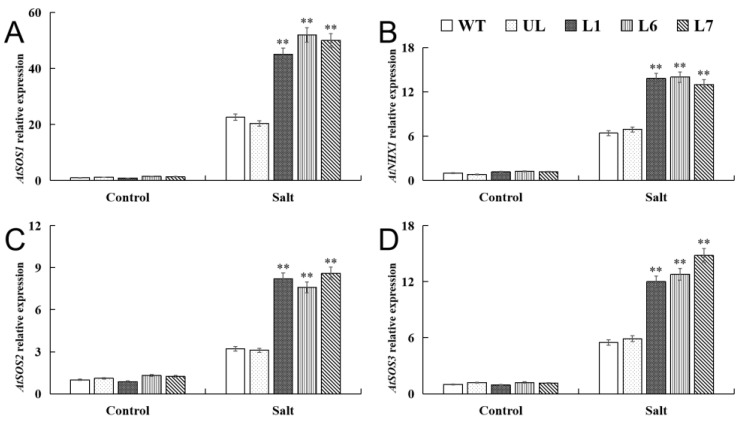
RT-qPCR detection of salinity tolerance-related gene expression in transgenic *A. thaliana* plants. L1, L6, and L7 are *A. thaliana* overexpression lines. Relative expression levels of (**A**) *AtSOS1*, (**B**) *AtNHX1*, (**C**) *AtSOS2*, and (**D**) *AtSOS3* in the WT, UL, and *VhMYB2*-OE lines (L1, L6, and L7). Data represent the average of three repetitions. Asterisks above columns indicate significant difference compared to WT (** *p* ≤ 0.01).

**Figure 9 ijms-24-10743-f009:**
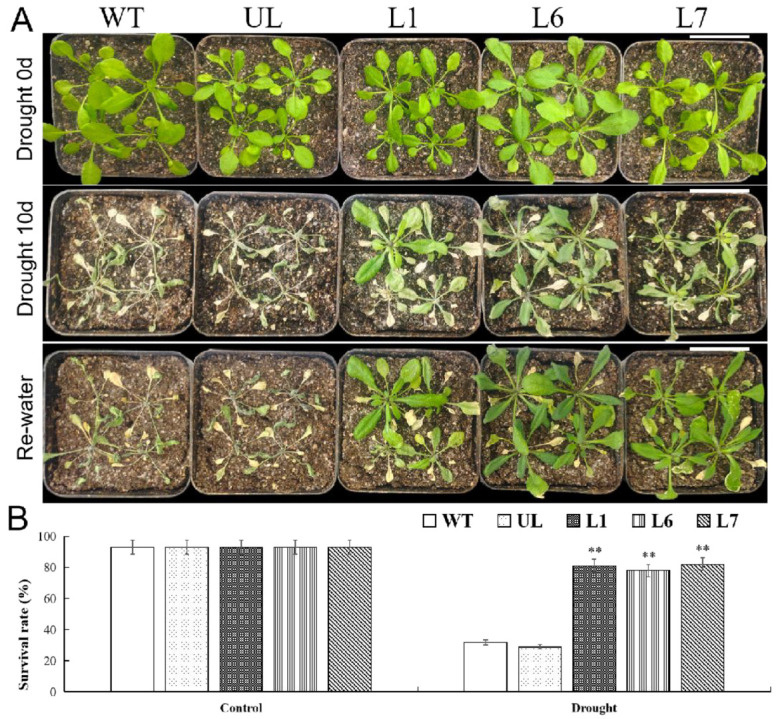
Drought tolerance of 35S::*VhMYB2* transgenic *A. thaliana* plants. (**A**) Phenotypes of WT, UL, and transgenic lines (L1, L6, and L7) with high expression. Scale white bar = 3.5 cm. (**B**) Survival rates of *A. thaliana* under drought stress. Asterisks above columns indicate significant differences compared to WT (** *p* ≤ 0.01).

**Figure 10 ijms-24-10743-f010:**
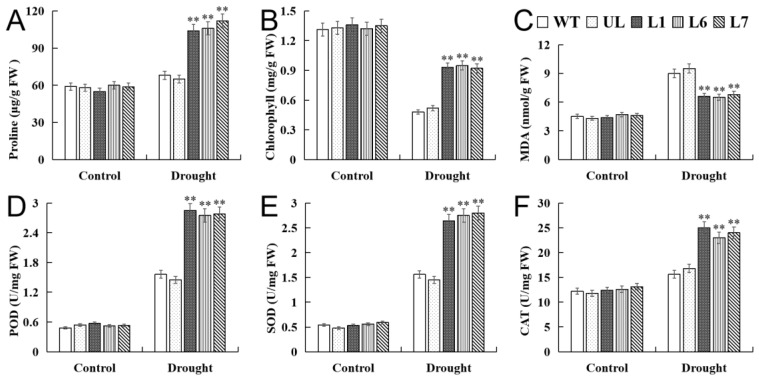
Analysis of proline (**A**), chlorophyll (**B**), MDA (**C**) POD (**D**), SOD (**E**), and CAT (**F**) in WT and 35S::*VhMYB2* transgenic *A. thaliana* plants. These were measured at 0 d and 14 d after drought treatment. Asterisks above the error bars indicate a significant difference between transgenic and WT lines (** *p* ≤ 0.01). The level of each index in WT was used as control.

**Figure 11 ijms-24-10743-f011:**
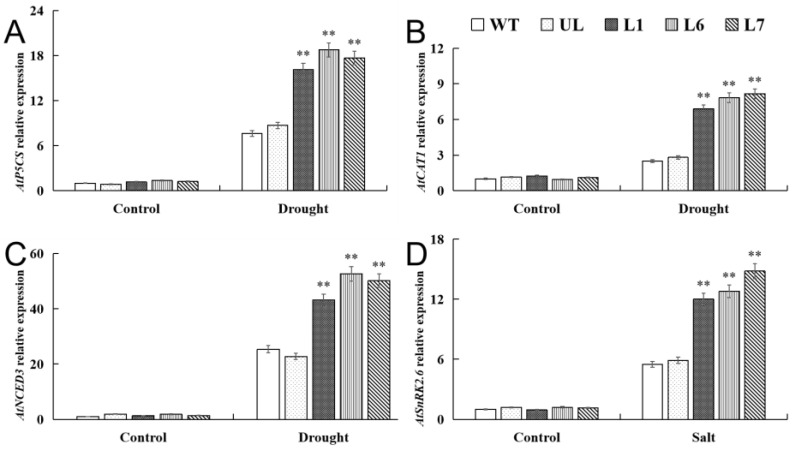
RT-qPCR detection of drought tolerance-related gene expression in transgenic *A. thaliana* plants. L1, L6, and L7 are *A. thaliana* overexpression lines. Relative expression levels of (**A**) *AtP5CS*, (**B**) *AtCAT1*, (**C**) *AtNCED3*, and (**D**) *AtSnRK2.6* in the WT, UL, and *VhMYB2*-OE lines (L1, L6, and L7). Data represent the average of three repetitions. Asterisks above columns indicate significant difference compared to WT (** *p* ≤ 0.01).

**Figure 12 ijms-24-10743-f012:**
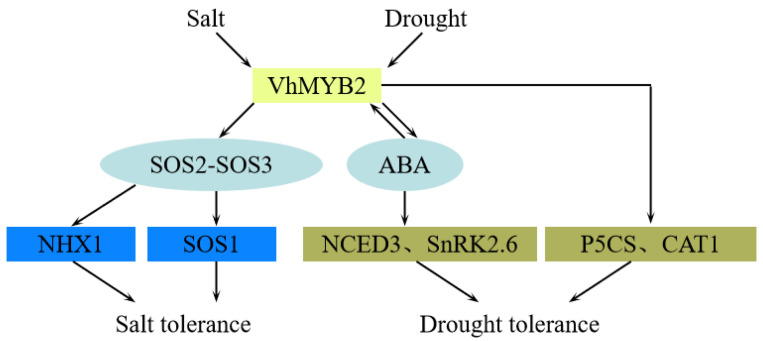
A potential model of *VhMYB2* response to salinity and drought stress.

## Data Availability

Not applicable.

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
