# Peer review of "Overexpression of a Grape MYB Transcription Factor Gene VhMYB2 Increases Salinity and Drought Tolerance in Arabidopsis thaliana"

_ijms, 2023, doi:10.3390/ijms241310743_

Round 1

Reviewer 1 Report

Summary:

The study investigates the role of the VhMYB2 gene, a transcription factor, in conferring salinity and drought tolerance in the 'Beta' grape rootstock. The authors cloned and characterized the VhMYB2 gene, determining its subcellular localization and expression pattern in different tissues. However, the experimental design and methodology do not provide sufficient evidence to establish a key role of VhMYB2 in regulating salinity and drought tolerance. Additionally, the overexpression of VhMYB2 in Arabidopsis thaliana may not represent a physiological condition in which a transcription factor could activate genes that are not activated under standard conditions. It would be interesting to evaluate if the overexpression of the homologous VvMYB2 gene from Vitis vinifera in Arabidopsis thaliana can elicit similar responses. The discussion lacks stimulating insights and fails to propose further investigations or possible avenues of research. It would be valuable to outline the authors' plans for future research, such as investigating the characteristics of VhMYB2 in the 'Beta' rootstock's genome, regulating its expression, or developing/selecting knockout mutants to demonstrate its crucial role in regulating the rootstock's resistance to abiotic stress.

General Concept Comments:

 The study addresses an important topic by investigating the molecular mechanisms underlying salinity and drought tolerance in grape rootstocks.

The identification of VhMYB2 as a potential transcription factor involved in stress response provides valuable insights into the genetic basis of stress tolerance.

The experimental design and methodology do not sufficiently demonstrate a key role of VhMYB2 in the regulation of salinity and drought tolerance.

Considering the overexpression of VhMYB2 in Arabidopsis thaliana, it would be beneficial to assess if the overexpression of the homologous VvMYB2 gene from Vitis vinifera can produce similar responses.

Specific Comments: 

Figure 12: While the figure presents a potential model of the role of VhMYB2 in salinity and drought stresses, it lacks a description of the elements involved in signal transduction that activate VhMYB2, while the crucial role of VhMYB2 in promoting the downstream expression of salinity and drought stress-related genes are not demontrate.

Overall, the study highlights the potential significance of VhMYB2 in enhancing salinity and drought tolerance in Arabidopsis thaliana, suggesting its potential as a target for improving stress resistance in grape cultivars. However, additional clarifications and improvements are needed to strengthen the presentation of the research findings.

Reviewer 2 Report

Article Overexpression of a Grape MYB Transcription Factor Gene VhMYB2 Increases Salinity and Drought Tolerance in Arabidopsis thaliana by authors Chuankun Ren, Zhenghao Li, Penghui Song, Yu Wang, Wanda Liu, Lihua Zhang, Xingguo Li, Wenhui Li, Deguo Han is a feature study reactions of transgenic Arabidopsis thaliana plants expressing MYB Transcription Factor Gene VhMYB2. The materials appear to be of fairly high quality and may be of interest to experts in the field.

However, the manuscript was not prepared according to the rules of the journal, and its evaluation presents a certain difficulty, since apparently the authors had problems with preparing the manuscript or this manuscript was being prepared for submission to a free printed journal. In addition, it is rather strange to present histograms in black and white in a color magazine. There is a duplication of numbers in the bibliography.

It's not very clear to me what is meant by "Based on the spatio-temporal expression of VhMYB2 gene under different abiotic stresses".

I would like to clarify what the authors mean by generation T1 - T3? although there is - it should be written, even if it seems obvious to the authors, since it is not obvious to me that T0 is not T1!

I do not see the statistics section and without it I cannot evaluate the histograms, under each histogram, the corresponding value must be indicated. It is possible to state that the localization of the protein in the nucleus based on the analysis of epidermal cells only, but this may be an artifact or due to tissue specificity.

I think that if the article is framed according to the rules and the inaccuracies of the design are eliminated, it can be printed.

Reviewer 3 Report

Unfortunately, this work "Overexpression of a Grape MYB Transcription Factor Gene VhMYB2 Increases Salinity and Drought Tolerance in Arabidopsis thaliana" did not make a very good impression on me. The work is framed rather sloppy. There is no line numbering in the manuscript itself, so working with the manuscript is rather difficult. There are no figures in the text of the manuscript at all. The work with the text is also difficult. Will all drawings be in Supplementary Materials? The figures are not signed, and the legends are displayed separately after the References, which also makes it difficult to work with the manuscript.

The title of the article refers to the study of Arabidopsis as a plant object, but, in fact, this study is carried out on three objects - Vitis, Nicotiana and Arabidopsis. Neither the Abstract nor the Conclusion shows this, and it is difficult for readers to understand. If the authors studied the physiological properties of transgenic Arabidopsis, then why did they evaluate the expression of VhMYB2 in Vitis hybrid? How is this related to Arabidopsis? Unclear! Why did the authors look at the localization of VhMYB2 in tobacco nuclei? After all, the authors obtained stably transformed Arabidopsis plants. Do the authors believe that the same result will be observed in Arabidopsis leaves? But this may not be the case. At least, this had to be shown experimentally on Arabidopsis. Why, by the way, are the cells of the epidermis, and not of the spongy parenchyma or meristem?

Abstract

…“we demonstrated VhMYB2 was a key transcription factor”… a controversial statement. It is better to say - among other transcription factors, it made a significant contribution .... (for example)

…”which was cloning in ‘Beta’rootstock”… better “was isolated from”…

…”Subcellular localization of VhMYB2 was located”… - In what plant?

…”VhMYB2 was predominantly expressed in mature leaf “… - In what plant?

Introduction

The last 6 lines are badly written. Needs to be rewritten. No need to write what was done. This should be moved to the Conclusion. And here you need to clearly write what the goal was set by the authors in this study. “Based on the spatio-temporal expression of VhMYB2 gene under different abiotic stresses”… is a completely doubtful statement based on the results of the study.

Results

The fifth line in the subsection of VhMYB2 Cloning and Phylogenetic Analysis – “VhMYB2 protein consisted of 20 amino acids” – is unclear.

Subsection “VhMYB2 was Localized onto Nucleus” – in nucleus? Already wrote - why do we need tobacco?

Subsection “Expression Level Analysis of VhMYB2 in ‘Beta’” – why is this analysis needed if the work is related to transgenic Arabidopsis?

Subsection “VhMYB2 in Transgenic A.thaliana Enhanced Salinity Tolerance” The abbreviation UL needs to be explained.

Conclusion

Written badly. The results need to be displayed more clearly. “The expression levels of VhMYB2 were induced by salinity and drought”… - in which plants?

My main complaint about this work concerns the results presented in Figures 7 and 10. They show that under stress (salt or drought) there was an increase in proline content, a decrease in MDA and an increase in POD, SOD and CAT activity. This would be true if the gene of interest (VhMYB2) were in the genetic construct under the control of an inducible promoter. And please note that in the absence of stress (salt, drought), this promoter would not be induced, and, accordingly, the expression of downstream genes associated with the stress response would not occur. Just like the authors in Fig. 7 and 10. However, an expression cassette with the CaMV 35S promoter was used to stably transform Arabidopsis. I draw your attention to the fact that this promoter is constitutive and works even in the absence of stress. Accordingly, the VhMYB2 transcription factor must necessarily induce downstream genes, regardless of whether stress occurs or not. And in Figures 7 and 10, in the absence of stress, there is no proline induction, an increase in POD, SOD and CAT activity, although it should be. How can it be? Authors must explain this in the Discussion section, otherwise these results are questionable. This manuscript needs serious revision.

Reviewer 4 Report

In real conditions, a plant can be affected by more than one factor, but several at once. In this manuscript, the effect of two factors on Arabidopsis at once was considered: drought and salinity, and options for increasing the plant's tolerance to them. the study was carried out at a high level using many modern methods. However, there are several remarks.

In the Introduction section there is a recap with references (28-30) to Myb2 involvement in abiotic stress. But this has been discussed in detail above.

Section 2.3. Line 5 - We are talking about an amino acid, not a nucleic acid

SANT - what is this structure? Is it an amino acid motif? specify

Section 2.3. From the data in Fig. 5A does not imply that the main function of VhMyb2 is nutrient transport

give a more detailed description of the methods for determining activities

Include pictures in text with captions. There was an imposition of the numbering of references on the captions to the figures

references are double numbered

Reviewer 5 Report

The authors of the manuscript titled “Overexpression of a Grape MYB Transcription Factor Gene VhMYB2 Increases Salinity and Drought Tolerance in Arabidopsis thaliana” In the current study, the authors investigated the potential roles of VhMYB2 in salinity and drought resistance.

General comments

Overall, the study is well-designed and presented in a good way.

Abstract

The authors are requested to add quantitative data on the studied parameters and mention that the role of the target gene was more prominent in salt stress or drought stress.

Introduction

This section is also well written but the authors are requested to delete the extra or irrelevant references because overall the number of references in the manuscript is more. In a research article, the number of references should not be more than 40 and here almost 80.  So, please the extra or irrelevant references from the manuscript.

Materials and Methods

This section is well-written.

Results

This section is well-written.

Discussion

This section is well-written but the authors are requested to delete the extra or irrelevant references.

Conclusion

The authors are requested to add that the role of the target gene was more prominent in drought or salinity stress.

Round 2

Reviewer 2 Report

The manuscript has been improved. The authors have done some work. It is desirable to improve the wording of the sentences in the Conclusion section.

Reviewer 3 Report

My comments are in file

Round 3

Reviewer 3 Report

The authors have corrected the remarks made, so I believe that the manuscript can be accepted in its present form.